# Association of sociodemographic characteristics with self-perceived access to COVID-19 information and adherence to preventive measures among migrant origin and general populations in Finland: a cross-sectional study

Natalia Skogberg  , Tyler Prinkey, Eero Lilja, Päivikki Koponen, Anu E Castaneda

Public Health and Welfare, Finnish Institute for Health and Welfare, Helsinki, Finland

**Correspondence to**
Dr Natalia Skogberg;
natalia.skogberg@thl.fi

## ABSTRACT

**Objectives** This study examines how access to COVID-19 information and adherence to preventive measures varies by sociodemographic characteristics, and whether the associations differ among the migrant origin and the general Finnish population. Additionally, the association of perceived access to information with adherence to preventive measures is examined.

**Design** Cross-sectional, population-based random sample.

**Background** Equity in access to information is crucial for securing individual well-being and successful management of a crisis at population level.

**Setting** Persons who have a residence permit in Finland.

**Participants** Migrant origin population constituted of persons aged 21–66 years born abroad, who took part in the Impact of the Coronavirus on the Wellbeing of the Foreign Born Population (MigCOVID) Survey conducted from October 2020 to February 2021 (n=3611). Participants in the FinHealth 2017 Follow-up Survey conducted within the same time frame, representing the general Finnish population, constituted the reference group (n=3490).

**Outcome measures** Self-perceived access to COVID-19 information, adherence to preventive measures.

**Results** Self-perceived access to information and adherence to preventive measures was overall high both among the migrant origin and the general population. Perceived adequate access to information was associated with living in Finland for 12 years or longer (OR 1.94, 95% CI 1.05–3.57) and excellent Finnish/Swedish language skills (OR 2.71, 95% CI 1.62–4.53) among the migrant origin population and with higher education (OR 3.56, 95% CI 1.49–8.55 for tertiary and OR 2.87, 95% CI 1.25–6.59 for secondary) among the general population. The association between examined sociodemographic characteristics with adherence to preventive measures varied by study group.

**Conclusions** Findings on the association of perceived access to information with language proficiency in official

## STRENGTHS AND LIMITATIONS OF THIS STUDY

⇒ Significant strengths of the current study are its population-based study design, representative sample and availability of the general population reference group.
⇒ Barriers to participation were reduced by making the survey material available in 18 languages both in electronic and paper-based format, as well as by providing with the option to respond via a telephone interview conducted by multilingual interviewers.
⇒ One limitation is that self-reported access to information does not provide information on the quality of the used sources or whether the information was understood and applied as intended.
⇒ There may be some report bias in the self-reported adherence to preventive measures due to social desirability. It may also be that respondents reported adherence to preventive measures, even though they may have not adhered to them in all situations.

languages highlight the need for rapid multilingual and simple language crisis communications. Findings also suggest that crisis communications and measures designed to influence health behaviours at population level may not be directly transferable if the aim is to influence health behaviours also among ethnically and culturally diverse populations.

## BACKGROUND

COVID-19 exposed the weakest spots in crisis preparedness, highlighting the evident lack of policies and practices that acknowledge increasing globalisation and population cultural diversity.[1] Vulnerabilities of migrant origin populations were reflected in significantly higher incidence of COVID-19 infections, higher risk for hospitalisations,

mortality rates, as well as higher economic, social and psychological impact of the pandemic compared with general populations.[2–4] Information guidance was crucial in management of the pandemic; however, crisis communications were challenged by ever-evolving information and changes in official guidelines.[5 6] From the onset of the pandemic, significant concerns were raised regarding equity in access to information among migrant origin populations. Difficulties in accessing up-to-date and reliable information occurred due to language barriers, lack of trust in the health system and gaps in health and digital literacy.[7–9] Inequities in access to information may lead to stress, lower awareness and, hence, also lower adherence to preventive measures.[7 10 11] In addition to impact on individual-level coping in a crisis, these may negatively impact overall crisis management and hence may jeopardise societal security.

While multilingual communications and community outreach programmes were launched in a number of countries to improve equity in access to information across different population groups,[12] practices across countries varied. During the first half year of the pandemic, only half of European countries had information on COVID-19 available in at least one foreign language, and only several countries had information in multiple foreign languages.[13] Significant efforts have been placed to develop multilingual COVID-19 communications also in Finland.[14] Finland is a country in the North of Europe, with the population of 5.6 million. Migration to Finland rapidly accelerated over the past decades from 0.5% of the total population in the 1990 to 8% in 2021.[15] In 2021, the largest migrant origin populations in Finland were persons born in the former Soviet Union (FSU), Estonia, Russian Federation, Sweden, Iraq, China and Somalia. A few weeks into the pandemic, the Finnish Institute for Health and Welfare (THL) introduced information on COVID-19 in 16 foreign languages. Language selection was later expanded by several additional languages. In addition to THL that produced crisis communications at the national level, regional health authorities and non-governmental organisations also produced multilingual communications. It was well acknowledged that simply publishing multilingual information is not sufficient, and particular efforts were taken to ensure effective information dissemination to the target group. Dissemination of national-level COVID-19 communications was implemented by THL's multilingual and multichannel communications task force in close collaboration with regional health and social welfare professionals, non-governmental organisations, religious community leaders and other key community representatives.[16]

Because multilingual COVID-19 communications were available with a slight delay, migrant origin populations in Finland initially perceived they did not receive sufficient information, but the situation improved over the course of the pandemic.[8] To date, studies exploring perceived access to information and adherence to preventive measures among migrant origin populations have been generally based on qualitative and non-representative samples.[7–9 17] The general population reference group is rarely available, which makes contextualisation of the findings challenging. There is also limited information on how access to information and adherence to preventive measures has varied by sociodemographic characteristics, and whether these differed between migrant origin and the general population. Furthermore, there is limited information on how perceived access to information has impacted behaviours in different population groups. This information is needed for evaluation of crisis response measures and for identifying groups at highest risk of vulnerabilities in crisis situations. These all significantly contribute to development of preparedness in future crises. The aim of the current study is to fill this gap in knowledge by using representative population-based data to examine how access to COVID-19 information and adherence to preventive measures varied by sociodemographic characteristics, and whether these differed between the migrant origin and the general Finnish populations. Additionally, the association of access to information with adherence to preventive measures is examined.

## METHODS

The present study is based on the Impact of the Coronavirus on the Wellbeing of the Foreign Born Population (MigCOVID) Survey. The survey was conducted between October 2020 and February 2021, coinciding with the second wave of the COVID-19 pandemic in Finland. A more detailed description of the survey can be found elsewhere.[4] Briefly, the MigCOVID Survey was a 2-year follow-up study to the Survey on Wellbeing among Foreign Born Population (FinMonik).[18] The original stratified random sample was drawn from the Finnish Population Register using the following selection criteria: age 18–64 years, both the person belonging to the sample and their parents were born outside of Finland and residence in Finland for at least 1 year. Following exclusion of persons declining further contact during the FinMonik Survey and who no longer lived in Finland, the updated sample of the MigCOVID Survey comprised of 5269 persons aged 20–66 years. In addition to including participants in the FinMonik Survey, a supplementary sample of 982 persons of Somali origin was drawn using simple random sampling and following the same selection criteria as the original sample. Following exclusion of overcoverage (n=112), the final sample of the MigCOVID Survey constituted 6139 persons. Altogether, 3668 persons took part in the MigCOVID Survey, with a participation rate of 60%. Since data for the general population reference group were available only for persons aged 21 years and older, n=57 persons below 21 years were excluded from the MigCOVID participants, resulting in use of data of n=3611 participants in the MigCOVID Survey in the current study.

Invitation letters were sent by mail with personal login information to the electronic questionnaire. Data

collection was supplemented with mailed paper questionnaires and telephone interviews. Invitation letters, information about the survey and questionnaires were available in 18 languages. Invitation letters and paper questionnaires were sent out to the participants in Finnish or Swedish, depending on which of these was marked as the preferred language of communication in the Finnish Population Register, and in their official mother tongue as marked in the register. If the questionnaire was not available in the register-based mother tongue of the participant, the post was sent in Finnish/Swedish only. Participants could themselves select the language in which they filled out the electronic questionnaire out of the available 18 languages. Telephone interviews were carried out by trained multilingual fieldwork personnel in 12 languages.

A reference group was formed from a subpopulation of the FinHealth 2017 Follow-up Study of corresponding age to the participants in the MigCOVID Survey. As the youngest participants in the FinHealth 2017 Follow-up Study were 21 years, the reference group of this study constitutes persons aged 21–66 years (n=3490 participated, participation rate 51%). The FinHealth 2017 Study had a nationally representative two-stage stratified cluster sample of adults living in Finland.[19] The study was conducted within the same time frame and following a similar protocol to the MigCOVID Survey.

### Patient and public involvement in research

This study does not involve patients. The public were not involved in the planning or design of the research. Data collection was implemented by trained multilingual fieldwork personnel. Findings have been disseminated to the public in multiple languages through the web pages of the survey and social media. Additionally, group discussions with representatives of different migrant origin populations have been conducted to disseminate and to hear feedback on the findings.

### Variable definitions

Information on date of birth, age, sex and length of residence in Finland was obtained from the Finnish Population Register. The MigCOVID Survey participants were grouped into six regions of origin based on their country of birth following a slightly modified United Nations area code grouping[20]: Russia or the FSU; Estonia; the rest of Europe, North America and Oceania; the Middle East and North Africa; the rest of Africa (later in the text as Africa); Asia or other (later in the text as Asia). In the latter group, 'other' constituted persons born in Latin America (n=96). Information on highest completed level of education in Finland or abroad, economic activity and Finnish/Swedish language skills (official languages of Finland) was gathered as self-report measures during the survey. Categorisation of the included sociodemographic characteristics can be found in table 1.

Self-perceived access to information was assessed in both studies with the question: Have your received adequate information on how to avoid getting infected with the coronavirus and how to prevent it from spreading? The answer options were: I have not received any information or the information I have received has been completely inadequate; I have received information, but I would have needed more; or I have received adequate information. The first two answer options were grouped.

Adherence to preventive measures was based on the measures recommended by THL at the point of finalisation of the questionnaire in September 2020. Participants were asked: Which measures have you taken to avoid getting infected with the coronavirus and prevent it from spreading: I wash my hands more frequently; I use sanitisers more frequently; I take care of hygiene when coughing (eg, coughing into a disposable tissue, not coughing into hands); I stay at home if I have flu symptoms (eg, cough, cold symptoms or sore throat); I wear a single-use mask or cloth mask during free time (when it is not possible to avoid contact with other people); I keep a safe distance from other people outside of the home; I do not shake hands with the people I meet. The answer options were: Yes, I follow the instruction/recommendation, and I do not follow the instruction/recommendation. The questions on washing hands, use of sanitisers and coughing hygiene were grouped into a joint variable 'improved hand and coughing hygiene'. Answers were dichotomised as 'yes' (yes to all three) and 'no' (no to at least one measure).

Participants were also asked if they have downloaded the Koronavilkku contact tracing app. In the MigCOVID Survey, the answer options were: yes; no, because the app is not available for my phone; no, because I don't know what it is; no, because the app is not available in the languages that I speak; no, for other reasons, what? The answers were dichotomised as yes/no. The open-ended answers were analysed using thematic grouping. In the FinHealth 2017 Follow-up Study, the answer options were: yes; no, because I don't have a suitable smartphone; no, for other reasons. These were also dichotomised as yes or no.

### Statistical analyses

The differences between groups were tested using logistic regression. P value based on Satterthwaite-adjusted F value, as well as OR and its 95% CI, was used to assess the differences. The model-adjusted proportions were calculated with predictive margins.[21] The stratification of the sample was accounted for in all analyses and finite population correction was applied.[22] The data were weighted using inverse probability weights to reduce the non-response bias and to take varying sampling probabilities into account.[23] The calculation of the inverse probability weights is described elsewhere.[4 19] All analyses were carried out with SAS V.9.4 and SUDAAN V.11.0.3 statistical software.

**Table 1** Descriptive characteristics of the study participants

| | General population N=3490 | Born abroad total N=3611 | Russia or former Soviet Union N=1115 | Estonia N=300 | Rest of Europe, North America and Oceania N=712 | Middle East and North Africa N=436 | Rest of Africa N=302 | Asia N=746 |
|---|---|---|---|---|---|---|---|---|
| | n (%) | n (%) | n (%) | n (%) | n (%) | n (%) | n (%) | n (%) |
| Men | 1604 (50.5) | 1614 (51.6) | 380 (39.0) | 110 (46.5) | 399 (64.7) | 286 (68.8) | 174 (54.9) | 265 (42.7) |
| Age (years) | | | | | | | | |
| 21–34 | 533 (29.3) | 1090 (34.7) | 251 (24.5) | 56 (25.0) | 179 (32.5) | 178 (49.7) | 156 (45.7) | 270 (38.1) |
| 35–49 | 1142 (33.6) | 1509 (41.7) | 412 (38.0) | 119 (39.7) | 339 (48.1) | 178 (36.2) | 122 (46.9) | 339 (43.0) |
| 50–66 | 1815 (37) | 1012 (23.6) | 452 (37.5) | 125 (35.3) | 194 (19.4) | 80 (14.1) | 24 (7.4) | 137 (18.9) |
| Education | | | | | | | | |
| Basic education or less | 273 (6.5) | 380 (13.2) | 57 (6.0) | 32 (16.0) | 55 (11.8) | 86 (16.8) | 56 (21.0) | 94 (14.2) |
| Secondary education | 1764 (49.7) | 1276 (46.1) | 435 (56.6) | 147 (59.2) | 210 (33.5) | 160 (50.8) | 117 (42.4) | 207 (36.2) |
| Tertiary education or higher | 1412 (43.8) | 1888 (40.8) | 609 (37.4) | 113 (24.8) | 440 (54.6) | 174 (32.4) | 119 (36.6) | 433 (49.6) |
| Economic activity | | | | | | | | |
| Employed full time/part-time | 2404 (70.7) | 2299 (65.7) | 699 (64.8) | 218 (70.9) | 504 (72.5) | 197 (51.6) | 172 (59.9) | 509 (69.7) |
| Unemployed, student or other | 1034 (29.3) | 1229 (34.3) | 396 (35.2) | 77 (29.1) | 195 (27.5) | 225 (48.4) | 119 (40.1) | 217 (30.3) |
| Length of residence in Finland (years) | | | | | | | | |
| 3–6.99 | N/A | 793 (17.9) | 163 (6.7) | 32 (6.5) | 134 (22.6) | 183 (31.8) | 46 (14.3) | 235 (24.0) |
| 7–11.99 | N/A | 1020 (28.7) | 273 (18.9) | 124 (43.4) | 200 (28.2) | 116 (30.4) | 58 (21.3) | 249 (31.6) |
| 12+ | N/A | 1798 (53.4) | 679 (74.3) | 144 (50.1) | 378 (49.2) | 137 (37.9) | 198 (64.3) | 262 (44.3) |
| Excellent Finnish/Swedish language proficiency | N/A | 1364 (35.4) | 480 (43.2) | 182 (52.1) | 273 (32.0) | 122 (29.1) | 163 (50.3) | 144 (17.8) |

N/A, not available.

## RESULTS

Descriptive characteristics of the study participants are presented in table 1. The proportion of men was lowest among persons born in Russia/FSU and highest among persons born in the Middle East and North Africa. Persons born in Africa were the youngest. The proportion of those with at least tertiary education was lowest among persons born in Estonia and highest among persons born in the rest of Europe, North America and Oceania. Employment was lowest among persons born in the Middle East and North Africa. Persons born in Russia/FSU had lived in Finland the longest and had the highest Finnish/Swedish language proficiency.

Perceived access to information and adherence to preventive measures among the study participants is presented in table 2. Migrant origin population reported receiving fully adequate COVID-19 information less frequently than the general population. Perceived access to information was lowest among persons born in Europe (excluding Russia/FSU, Estonia), North America and Oceania, the Middle East and North Africa, and Asia.

Persons born in Russia/FSU and Estonia reported lower, while persons born in the Middle East and Africa reported higher adherence to preventive measures compared with the general population. Migrant origin population reported downloading the Koronavilkku contact tracing app less frequently than the general population. The most common reasons mentioned for not downloading the app were not knowing about it; the app was unnecessary; not wanting to; the app would just increase stress; mistrust towards the authorities; technical difficulties; and that the app consumes too much phone memory (results not shown).

The association of sociodemographic characteristics with self-perceived access to information and adherence to COVID-19 preventive measures among the general population is presented in table 3. Higher education was positively associated with self-perceived sufficient access to information. Women were more likely than men to report adhering to all other preventive measures except for keeping a safety distance from others. Older age was positively associated with improved hand hygiene and use of a face mask. Additionally, persons in the oldest age group (50–66 years) were more likely to keep a safety distance from others but less likely to download the contact tracing app compared with the youngest (21–34 years) age group. Persons with tertiary education were more likely to report the use of a face mask than persons with basic education. Higher education and employment were also associated with downloading the contact tracing app.

The association of sociodemographic characteristics with self-perceived access to information and adherence to COVID-19 preventive measures among the migrant origin population is presented in table 4. Longer length of residence and excellent Finnish/Swedish skills were associated with perceived sufficient information on COVID-19. Women were generally more likely to adhere

to preventive measures than men. Older age was associated with a lower likelihood of reporting staying at home with flu symptoms and downloading the Koronavilkku contact tracing app, while tertiary education was associated with a greater likelihood of downloading the Koronavilkku app. Secondary education was associated with being less likely, whereas excellent Finnish/Swedish skills were associated with being more likely to avoid shaking hands with others. Those who have lived in Finland for 7–12 years were less likely to use a face mask than those who have lived in Finland for a shorter period.

The association of self-perceived access to information with adherence to preventive measures is presented in table 5. When the migrant origin population was examined as one group, the association was statistically significant for keeping a safety distance from others only. When examined by region of origin, statistically significant differences emerged in the population born in Africa in case of all preventive measures except for downloading the Koronavilkku app. Because of widened CIs in the subgroup analyses, interaction analyses were run to examine whether the associations for this group really differed from other regional groups. The p values for the differences of the Africa group versus other groups combined were p=0.101 for staying at home with flu symptoms; p=0.018 for following improved hand and coughing hygiene; p=0.074 for using a face mask; p=0.255 for keeping a safety distance from others; p=0.034 for avoiding shaking hands with others; and p=0.504 for downloading the Koronavilkku app. Based on these, it appears that the association of perceived access to information and following preventive measures was different for this regional group.

## DISCUSSION

Perceived access to information and self-reported adherence to preventive measures were high both in the migrant origin and the general population. Higher education in the general population and longer length of residence and excellent Finnish/Swedish language skills in the migrant origin population were associated with higher perceived access to information. In the general population, being a woman, older age and higher education were positively associated with adherence to preventive measures. While women reported higher adherence to preventive measures also in the migrant origin population, the association of other sociodemographic characteristics was less consistent than in the general population. Perceived sufficient access to information was associated with keeping a safety distance from others in the migrant origin population. In the analyses by region of origin, perceived access to information was associated with adherence to all preventive measures except for downloading the Koronavilkku contact tracing app among persons born in Africa.

Despite overall high perceived access to information, migrant origin populations reported they would have

**Table 2** Perceived access to information and self-reported adherence to preventive measures among the study population†

| | General population N=3490 | Born abroad total N=3611 | Russia or former Soviet Union N=1115 | Estonia N=300 | Rest of Europe, North America and Oceania N=712 | Middle East and North Africa N=436 | Rest of Africa N=302 | Asia N=746 |
|---|---|---|---|---|---|---|---|---|
| | % (95% CI) | % (95% CI) | % (95% CI) | % (95% CI) | % (95% CI) | % (95% CI) | % (95% CI) | % (95% CI) |
| **Access to information** | | | | | | | | |
| Received adequate information | 97.1 (95.7–98.0) | 93.5 (92.0–94.8)*** | 96.1 (94.3–97.4) | 97.4 (94.0–98.9) | 91.4 (87.0–94.5)*** | 91.8 (86.6–95.1)** | 93.9 (88.8–96.7) | 91.4 (86.8–94.5)*** |
| Received, but would have needed more | 2.5 (1.6–3.9) | 5.3 (4.2–6.7)*** | 2.8 (1.8–4.3) | 1.9 (0.7–5.5) | 6.2 (3.8–9.9)** | 7.8 (4.6–13.0)** | 5.4 (2.7–10.6) | 7.5 (4.8–11.6)*** |
| No information/ completely inadequate | 0.4 (0.2–0.8) | 1.1 (0.6–2.0)*** | 1.1 (0.5–2.4)* | 0.6 (0.2–2.4) | 2.4 (0.9–6.2)** | 0.4 (0.1–1.1) | 0.7 (0.3–1.9) | 1.1 (0.2–5.6) |
| **Following preventive measures** | | | | | | | | |
| Staying home with flu symptoms | 96.1 (94.6–97.2) | 96.4 (95.1–97.4) | 94.2 (91.4–96.1) | 92.1 (85.2–95.9) | 96.4 (92.8–98.2) | 99.2 (98.1–99.7)** | 99.0 (97.6–99.6)** | 98.2 (95.2–99.3) |
| Improved hand and coughing hygiene | 92.2 (90.3–93.7) | 92.8 (91.0–94.2) | 85.3 (81.2–88.7)*** | 84.4 (75.7–90.4)* | 96.3 (92.7–98.1)* | 95.7 (91.4–97.9) | 98.7 (96.9–99.5)*** | 96.2 (92.4–98.1) |
| Use of face mask | 81.9 (79.6–83.9) | 81.5 (79.0–83.7) | 69.5 (64.1–74.4)*** | 65.0 (55.5–73.5)*** | 84.2 (78.5–88.5) | 88.9 (83.5–92.7)* | 93.7 (90.4–96.0)*** | 89.2 (84.1–92.9)*** |
| Keeping a safety distance | 94.6 (93.0–95.8) | 93.5 (91.6–95.0) | 92.0 (88.5–94.6) | 87.1 (78.1–92.8)** | 92.6 (86.6–96.0) | 96.8 (93.0–98.6) | 97.3 (95.3–98.5)* | 94.9 (90.2–97.4) |
| Avoiding shaking hands with others | 98.8 (98.3–99.2) | 93.2 (91.5–94.6)*** | 89.7 (86.2–92.3)*** | 84.7 (75.5–90.9)*** | 95.8 (92.5–97.7)*** | 95.4 (92.2–97.4)*** | 95.7 (92.5–97.6)*** | 96.2 (92.0–98.3)** |
| Downloaded Koronavilkku contact tracing app | 62.4 (60.1–64.7) | 44.1 (41.2–47.2)*** | 33.5 (28.4–39.0)*** | 23.1 (16.6–31.2)*** | 53.5 (46.6–60.3)* | 51.4 (43.7–59.1)* | 50.8 (40.2–61.4)* | 52.8 (46.0–59.4)** |

*P<0.05; **p<0.01; ***p<0.001, in comparison with the general population.
†Adjusted for age and sex.

**Table 3** The association of sociodemographic characteristics with perceived adequate access to information and adherence to COVID-19-related preventive measures among the general population

| | Perceived access to information OR (95% CI) | Staying home with flu symptoms OR (95% CI) | Improved hand and coughing hygiene OR (95% CI) | Use of face mask OR (95% CI) | Keeping a safety distance from others OR (95% CI) | Avoiding shaking hands with others OR (95% CI) | Downloaded Koronavilkku app OR (95% CI) |
|---|---|---|---|---|---|---|---|
| Sex† | | | | | | | |
| Men | 1.00 | 1.00 | 1.00 | 1.00 | 1.00 | 1.00 | 1.00 |
| Women | 0.75 (0.36–1.55) | 2.53 (1.14–5.59)* | 2.43 (1.58–3.74)*** | 3.30 (2.49–4.38)*** | 1.17 (0.68–2.02) | 2.46 (1.16–5.19)* | 1.64 (1.36–1.98)*** |
| Age‡ (years) | | | | | | | |
| 21–34 | 1.00 | 1.00 | 1.00 | 1.00 | 1.00 | 1.00 | 1.00 |
| 35–49 | 1.51 (0.59–3.89) | 1.35 (0.61–3.00) | 1.92 (1.11–3.32)* | 1.50 (1.03–2.21)* | 1.24 (0.65–2.39) | 2.17 (0.90–5.25) | 0.79 (0.58–1.07) |
| 50–66 | 2.08 (0.80–5.36) | 2.21 (0.99–4.91) | 2.08 (1.23–3.52)** | 2.05 (1.41–2.97)*** | 2.46 (1.27–4.79)** | 1.96 (0.86–4.45) | 0.61 (0.45–0.81)*** |
| Education§ | | | | | | | |
| Basic education or less | 1.00 | 1.00 | 1.00 | 1.00 | 1.00 | 1.00 | 1.00 |
| Secondary education | 2.87 (1.25–6.59)* | 1.13 (0.48–2.69) | 1.12 (0.37–3.36) | 1.28 (0.88–1.87) | 0.94 (0.43–2.04) | 1.12 (0.37–3.36) | 2.13 (1.58–2.87)*** |
| Tertiary education or higher | 3.56 (1.49–8.55)** | 1.51 (0.61–3.75) | 1.19 (0.37–3.86) | 2.20 (1.41–3.44)*** | 1.10 (0.49–2.47) | 1.19 (0.37–3.86) | 4.48 (3.22–6.24)*** |
| Economic activity§ | | | | | | | |
| Unemployed, student or other | 1.00 | 1.00 | 1.00 | 1.00 | 1.00 | 1.00 | 1.00 |
| Employed full time/part-time | 0.88 (0.37–2.09) | 1.81 (0.86–3.83) | 0.89 (0.40–1.96) | 0.93 (0.66–1.30) | 0.80 (0.40–1.59) | 0.89 (0.40–1.96) | 0.63 (0.50–0.79)*** |

*P<0.05; **p<0.01; ***p<0.001.
†Adjusted for age.
‡Adjusted for sex.
§Adjusted for age and sex.

**Table 4** The association of sociodemographic characteristics with perceived adequate access to information and adherence to preventive measures

| | Perceived access to information OR (95% CI) | Staying home with flu symptoms OR (95% CI) | Improved hand and coughing hygiene OR (95% CI) | Use of face mask OR (95% CI) | Keeping a safety distance from others OR (95% CI) | Avoiding shaking hands with others OR (95% CI) | Downloaded Koronavilkku app OR (95% CI) |
|---|---|---|---|---|---|---|---|
| Sex† | | | | | | | |
| Men | 1.00 | 1.00 | 1.00 | 1.00 | 1.00 | 1.00 | 1.00 |
| Women | 1.22 (0.79–1.89) | 1.07 (0.60–1.93) | 2.10 (1.33–3.31)** | 1.70 (1.23–2.33)** | 2.12 (1.26–3.56)** | 2.41 (1.48–3.92)*** | 1.08 (0.84–1.39) |
| Age‡ (years) | | | | | | | |
| 21–34 | 1.00 | 1.00 | 1.00 | 1.00 | 1.00 | 1.00 | 1.00 |
| 35–49 | 0.94 (0.57–1.56) | 0.25 (0.14–0.47)*** | 1.01 (0.58–1.76) | 1.10 (0.76–1.59) | 1.36 (0.73–2.52) | 1.06 (0.62–1.82) | 0.61 (0.46–0.82)** |
| 50–66 | 1.39 (0.73–2.66) | 0.24 (0.12–0.48)*** | 0.93 (0.51–1.69) | 1.30 (0.86–1.97) | 2.09 (0.99–4.40) | 1.29 (0.68–2.43) | 0.35 (0.25–0.49)*** |
| Education§ | | | | | | | |
| Basic education or less | 1.00 | 1.00 | 1.00 | 1.00 | 1.00 | 1.00 | 1.00 |
| Secondary education | 0.99 (0.50–1.94) | 0.58 (0.14–2.40) | 0.57 (0.22–1.46) | 0.73 (0.42–1.25) | 0.87 (0.36–2.10) | 0.38 (0.15–0.95)* | 0.94 (0.63–1.41) |
| Tertiary education or higher | 0.66 (0.34–1.27) | 0.82 (0.19–3.56) | 1.25 (0.49–3.20) | 1.08 (0.62–1.88) | 0.86 (0.36–2.07) | 1.09 (0.42–2.86) | 1.72 (1.15–2.58)** |
| Economic activity§ | | | | | | | |
| Unemployed, student or other | 1.00 | 1.00 | 1.00 | 1.00 | 1.00 | 1.00 | 1.00 |
| Employed full time/part-time | 0.69 (0.43–1.13) | 1.18 (0.63–2.24) | 1.21 (0.76–1.94) | 1.11 (0.79–1.56) | 1.58 (0.87–2.87) | 0.87 (0.54–1.38) | 0.92 (0.71–1.20) |
| Length of residence in Finland§ (years) | | | | | | | |
| 3–6.99 | 1.00 | 1.00 | 1.00 | 1.00 | 1.00 | 1.00 | 1.00 |
| 7–11.99 | 1.15 (0.63–2.11) | 0.93 (0.30–2.84) | 0.50 (0.23–1.09) | 0.45 (0.27–0.75)** | 1.21 (0.52–2.82) | 0.68 (0.34–1.38) | 0.92 (0.71–1.20) |
| 12+ | 1.94 (1.05–3.57)* | 0.85 (0.31–2.30) | 0.50 (0.25–1.00) | 0.62 (0.38–1.01) | 1.35 (0.64–2.84) | 1.05 (0.56–1.97) | 0.92 (0.71–1.20) |
| Finnish/Swedish proficiency§ | | | | | | | |
| Intermediate/beginner or less | 1.00 | 1.00 | 1.00 | 1.00 | 1.00 | 1.00 | 1.00 |
| Excellent | 2.71 (1.62–4.53)*** | 0.98 (0.51–1.86) | 0.83 (0.52–1.30) | 1.04 (0.76–1.44) | 1.28 (0.72–2.26) | 1.73 (1.06–2.84)* | 1.13 (0.87–1.47) |

*P<0.05; **p<0.01; ***p<0.001.
†Adjusted for age.
‡Adjusted for sex.
§Adjusted for age and sex.

**Table 5** The association of adequate perceived access to information with following preventive measures by region of origin†

| | Staying home with flu symptoms OR (95% CI) | Following improved hand and coughing hygiene OR (95% CI) | Using a face mask OR (95% CI) | Keeping a safety distance from others OR (95% CI) | Avoiding shaking hands with others OR (95% CI) | Downloading the Koronavilkku app OR (95% CI) |
|---|---|---|---|---|---|---|
| General population | 0.58 (0.08–4.31) | 0.72 (0.23–2.24) | 0.95 (0.46–1.97) | 0.99 (0.27–3.59) | 1.20 (0.16–9.00) | 1.78 (0.90–3.52) |
| Born abroad total | 1.58 (0.60–4.16) | 0.73 (0.37–1.43) | 1.49 (0.88–2.53) | 2.54 (1.15–5.61)* | 1.53 (0.71–3.33) | 1.03 (0.63–1.69) |
| Russia or former Soviet Union | 1.01 (0.27–3.74) | 1.25 (0.46–3.39) | 2.87 (1.35–6.11) | 2.51 (0.80–7.90) | 1.57 (0.46–5.37) | 2.05 (0.87–4.81) |
| Estonia | 3.19 (0.44–23.28) | 0.25 (0.04–1.77) | 0.46 (0.08–2.62) | 4.56 (0.56–37.26) | 1.01 (0.15–6.95) | 1.18 (0.22–6.37) |
| Rest of Europe, North America and Oceania | 2.86 (0.46–17.83) | 0.74 (0.12–4.43) | 1.94 (0.57–6.65) | 4.22 (0.87–20.40) | 6.36 (1.42–28.51)* | 1.38 (0.49–3.85) |
| Middle East and North Africa | 0.54 (0.09–3.41) | 1.46 (0.15–14.60) | 1.98 (0.56–6.97) | 0.61 (0.10–3.77) | 0.87 (0.17–4.44) | 1.39 (0.45–4.29) |
| Rest of Africa | 8.29 (1.76–39.01)** | 15.06 (1.19–190.10)* | 6.13 (1.73–21.76)** | 6.63 (1.70–25.86)** | 15.50 (3.44–69.76)*** | 0.55 (0.13–2.27) |
| Asia | 0.40 (0.04–3.93) | 0.40 (0.09–1.82) | 1.56 (0.36–6.71) | 2.49 (0.30–20.43) | 0.13 (0.01–1.18) | 1.08 (0.37–3.14) |

*P<0.05; **p<0.01; ***p<0.001.
†Adjusted for age and sex; the reference group are those who perceived they did not receive fully adequate information on COVID-19 and related restrictive measures.

needed more information on COVID-19 more frequently than the general population. Furthermore, some regional group differences were observed, which may be attributable to the observed lower Finnish/Swedish language proficiency and shorter length of residence in some groups. Findings on the association of proficiency in Finnish/Swedish with perceived access to information are consistent with findings of a qualitative study conducted among Russian, Arabic and Somali-speaking populations at the start of the COVID-19 pandemic in Finland.[8] In the current study, perceived access to information was poorer in those who spoke Finnish/Swedish at intermediate level. This may reflect the complexity of crisis communications. Those who do not have sufficient Finnish/Swedish skills, and for whom official information on the Finnish context was not available in own language may be at a particular risk of not receiving sufficient information. While majority (91%) of migrant origin persons reported using Finnish media, use of international media (85%) and social media (82%), which may not necessarily convey accurate information of relevance to the Finnish context, were also frequently reported.[16] Future studies should explore the association of used sources with perceived access to information.

The observed association between length of residence with perceived access to information may be related to higher likelihood of attaining better Finnish/Swedish language proficiency with longer length of residence in Finland. It may also reflect accumulation of better general knowledge on where to source relevant information. The observed association of higher education with higher perceived access to information among the general population in this study is consistent with previous studies conducted among general populations.[24] Persons with higher education may be more skilled in seeking health-related information and may have better health literacy.[25]

Self-reported adherence to preventive measures was overall high both among the general and migrant origin populations in the current study, which is consistent with findings from Norway[16] and Switzerland,[26] but contradictory with findings from France.[27] Partially contradictory findings may be attributable to the fact that the study in France focused specifically on a disadvantaged migrant origin group from one specific region of origin, whereas the other studies included more heterogenous populations. In the current study, some differences in adherence to preventive measures were observed by region of origin. Lower adherence to preventive measures among persons born in Russia/FSU and Estonia compared with other groups may partially reflect behavioural patterns and official recommendations in the country of origin of participants. In Russia and Estonia, adherence to preventive measures has been reportedly influenced by the authorities' ambivalent and dismissive attitudes and the citizens' perceived lack of support.[28 29] In contrast, persons born in the Middle East and North Africa and the rest of Africa reported very high adherence to all preventive measures in this study, except for downloading

the contact tracing app. It is possible that these groups over-reported adherence to preventive measures due to social desirability. However, persons originating from the Middle East and Africa also had the highest incidence of COVID-19 in Finland,[30] therefore it is possible that these groups were more motivated to follow recommendations due to higher perceived risk of COVID-19 infection.

Women reported adherence to preventive measures more often than men both in the general and migrant origin populations, which is in consistency with previous studies.[24] Also in other areas of health and health-related behaviours, women generally have healthier habits than men.[18 31] Findings of a positive association of older age with improved hand and coughing hygiene, use of face mask and keeping a safety distance from others in the general population are also in consistency with previous studies.[24 32] In contrast, among the migrant origin population, older age was associated with being less likely to stay at home with flu symptoms. The reasons for this should be explored in future studies. Working conditions may also be associated with adherence to preventive measures. Less than a third of participants in the MigCOVID Survey who were working or training reported they were able to work remotely, 72% were able to take care of their hand hygiene at work and 56% were able to keep a safety distance from others.[16] Future studies should also explore the association of working conditions, such as ability to work remotely and avoid close contact with others, as well as the association of job security, with adherence to preventive measures.

Migrant origin populations reported downloading the contact tracing app less frequently than the general population, and the inverse association of older age with downloading the appwas observed. This is in consistency with previous studies and has been suggested to be related to lower digital competency among older adults.[33 34] Some of the participants indicated distrust towards authorities as a reason for not downloading the app, which is a concern raised among migrant origin populations also in previous studies.[35] Higher education was associated with higher likelihood of downloading the contact tracing app both among the general and migrant origin populations. The observed differences in how sociodemographic characteristics were associated with adherence to preventive measures in migrant origin population compared with the general population point to a need for tailoring population-level measures for them to be effective in influencing behaviours of diverse population groups.

In the current study, little association between perceived access to information and adherence to preventive measures was observed. Both perceived access to information and self-reported adherence to preventive measures were high, which may explain lack of the observed associations. However, when examined by region of origin, the association of adequate access to information and adherence to preventive measures was particularly observed among persons born in Africa. A large proportion of persons belonging to this group were of Somali

origin, which is a very close-knit community. Over the course of COVID-19, Somali origin community representatives and non-governmental associations were very active in distributing information within the community. Also the Somali-speaking television channel was active in disseminating information on COVID-19. Other possible reasons for the observed association should be explored in further studies. During the COVID-19 pandemic in Finland, social listening was introduced to gather signals from diverse population groups on their perceptions and needs related to the crisis.[36] Multilingual crisis communications were developed in close collaboration with regional health and welfare professionals, key community representatives and migrant origin populations. In addition to these, COVID-19 incidence was regularly followed and the MigCOVID Survey, which constitutes the basis for this study, was launched to gather population-based data on experiences during COVID-19 among migrant origin populations. When further developing future crisis preparedness, it may be worthwhile to consider use of additional systematic theory-driven tools, such as the framework for evaluating the effectiveness and sustainability of behaviourally and culturally informed interventions in complex settings, recently launched by WHO.[37]

Significant strengths of the current study are its population-based study design, representative sample and availability of the general population reference group. The survey was available in 18 of the most commonly spoken languages in Finland, reducing language barriers for majority of the participants. Participation barriers were further reduced through provision of the questionnaire in both electronic and paper-based format, as well as by providing an alternative to respond via a telephone interview. Some limitations also need to be addressed. Self-reported access to information does not provide information on the quality of the used sources or whether the information was understood and applied as intended. There may be some report bias in the self-reported adherence to preventive measures due to social desirability. It may also be that respondents reported adherence to preventive measures, even though they may have not adhered to them in all situations. Measured access to information, that may have reduced potential response bias, was not included in the current survey due to restrictions in the length of the questionnaire. However, when possible, research designing similar surveys in the future is encouraged to consider also including measured access to information.

When conducting surveys in multiple languages, there is a risk that the consistency of the items across the language versions may be jeopardised and that the translation process may alter the meaning of the survey items. In the current study, the translations were made by professional interpreters and have been reviewed by persons speaking the language in question, cross-checking them with the original questionnaire. Additionally, the multilingual interviewers, who were trained in the methodology of conducting surveys, importance of standardisation and

why the different questions were being asked, have also reviewed the questionnaires both from the perspective of consistency and conservation of the intended meaning. While it is still possible that some issues remained unnoticed, it is anticipated that these measures considerably reduced the problem of systematic misinterpretations related to translation error or the intended meaning of the items.

Considerable measures were taken to increase participation rate; however, the 18 languages in which the questionnaire was available did not exhaustively cover all of the different languages spoken by the participants. Those who did not participate may have had poorer literacy skills, poorer command of Finnish/Swedish and may have been likely to speak the languages in which the survey was provided. The languages selected in the survey covered the most commonly spoken languages in Finland. These broadly speaking (but not exactly) corresponded to the languages in which multilingual communications were produced in Finland. Therefore, it is possible that those who did not participate may have been at a greater risk of not receiving sufficient information. Furthermore, those who did not participate may have been less interested in their health and hence less likely to seek information or adhere to preventive measures. Non-response bias was addressed in this study through application of inverse probability weights.[23] These were based on register information on sociodemographics, and for the participants in the FinMonik sample, also earlier data on socioeconomic position and type of municipality of residence.[4] While the sample size and participation rate were substantial compared with other surveys conducted among migrant origin population during COVID-19, not all of the analyses could have been performed by region of origin.

In conclusion, this study provided evidence on importance of multilingual communications in securing equity in access to information among migrant origin populations in crisis situations. While policy and practice acknowledging the increasing cultural and linguistic diversity of populations has been evidently lacking prior to the pandemic, a number of countries have developed measures to improve access to up-to-date crisis communications in multiple languages during COVID-19. These measures need to be carefully documented and synthesised to identify best practices and to encourage mutual learning, as well as to facilitate better preparedness in future crises. Population diversity is a global trend that is accelerating rapidly. Therefore, it is imperative that multilingual and crisis communications are included in crisis preparedness plans. Countries must be ready to produce crisis communications in multiple languages without a significant time lag from the onset of a crisis to ensure adequate access to information among diverse population groups. Findings on differences in sociodemographic characteristics associated with access to information and adherence to preventive measures across migrant origin populations and also in comparison with the general population highlight the need for acknowledging population's linguistic and cultural diversity. It is imperative to apply comprehensive theory-driven monitoring and impact assessment measures to analyse the needs, perceptions and attitudes of diverse population groups during crisis situations to gain insights on drivers of behaviours, as well as to identify most effective strategies for influencing behaviours.

**Acknowledgements** The authors acknowledge all of the participants in the MigCOVID Survey and the Health 2017 Study, fieldwork personnel, as well as the experts involved in planning and data management of the studies.

**Contributors** NS conceptualised the paper. NS drafted the manuscript with assistance of TP. TP conducted the literature review. EL conducted the statistical analyses. PK and AC have provided their critical comments for developing the manuscript. All authors have participated in interpreting the results and reviewing drafts. All authors have read and approved the manuscript sent for publication. All authors accept full responsibility for the finished work, had access to the data, and controlled the decision to publish.

**Funding** The Impact of the Coronavirus on the Wellbeing of the Foreign Born Population (MigCOVID) Survey and the FinHealth 2017 Follow-up Study were funded from the Finnish parliament's additional budget allocated for COVID-19 research. NS and EL received funding from the Coping of disabled persons and persons of migrant origin in exceptional and crisis situations—Building the future based on experiences during COVID-19 (Building the Future) project supported by the European Social Fund (grant number: S22389). TP was funded by the MigCOVID Survey.

**Disclaimer** The funding bodies had no role in study design; in the collection, analysis and interpretation of data; in the writing of the report; or in the decision to submit the article for publication.

**Competing interests** None declared.

**Patient and public involvement** Patients and/or the public were not involved in the design, or conduct, or reporting, or dissemination plans of this research.

**Patient consent for publication** Obtained.

**Ethics approval** All participants were made aware of the voluntary nature of participation in the surveys, and that by participating they permitted the use of their personal information according to the data protection notification on handling personal data. The MigCOVID Survey received ethical permission from the Ethical Committee of THL (THL/4061/6.02.01/2020). The FinHealth 2017 Follow-up Study received ethical permission from the Ethics Committee II of the Helsinki and Uusimaa Hospital Region (HUS/2391/2020).

**Provenance and peer review** Not commissioned; externally peer reviewed.

**Data availability statement** Data are available upon reasonable request. The MigCOVID Survey data can be shared with other researchers within EU and ETA countries following an accepted research plan. More information: thl.fi/migcovid.

**ORCID iD**
Natalia Skogberg http://orcid.org/0000-0002-8829-4159

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
