## [Reviewer comments · BMJ Open]

ARTICLE DETAILS

TITLE (PROVISIONAL)	The association of sociodemographic characteristics with self-perceived access to Covid-19 information and adherence to preventive measures among migrant origin and general populations in Finland: a cross-sectional study
AUTHORS	Skogberg, N.; Prinkey, Tyler; Lilja, Eero; Koponen, Päivikki; Castaneda, Anu

VERSION 1 – REVIEW

REVIEWER	Jeffry Oktavianus City University of Hong Kong, Centre for Communication Research
REVIEW RETURNED	12-Nov-2022

GENERAL COMMENTS	Thank you for the opportunity to read the article comparing the access to COVID-19 information and adherence to preventive measures between migrant and general populations. The topic is timely and important, especially since migrant communities have been seen as more vulnerable during the health crisis. However, I have some minor recommendations and questions. 1. More background information about Finland and the migrant population will be helpful for the readers and can strengthen the importance of the study.2. The questionnaires were available in 18 languages. How did the authors ensure the consistency of the items across languages and that the translation process did not alter the meaning of the survey items?3. How did the authors develop the measurement for the perceived access to COVID-19 information? Also, why did the authors choose to measure the perceived access instead of the actual access to the information?4. Page 13, Line 19, should be 'perceived.'
---

REVIEWER	Marte Kjøllestad Norwegian Institute of Public Health
REVIEW RETURNED	21-Dec-2022

GENERAL COMMENTS	The association of sociodemographic characteristics with access to Covid-19 information and adherence to preventive measures among migrant origin and general populations in Finland: a cross-sectional study The study assesses self-perceived access to COVID-related information and adherence to preventive measures among immigrants and non-immigrants in Finland. The authors report their survey and the results in an organized and reasonable way, and gives valuable input to the planning of future pandemics and crises. I have only a few comments to consider, outlined below
--

	Methods Page 5, line 14; it is not clear what is FinMonik? Line 17: "In addition to including participants in the MigCOVID Survey, a supplementary sample of 982 persons of Somali origin was drawn using simple random sampling and following the same selection criteria as the original sample". This is not clear, is not the supplementary sample part of MigCOVID/this study? Russia or the former Soviet Union (FSU); Estonia; the rest of Europe, North America, and Oceania; the Middle East and North Africa; the rest of Africa; Asia or other (later in the text as Asia) What is "or other"? Please specify. In which group are immigrants from Latin America? Suggest to specify that "rest of Africa" will be later called "Africa", so you do not have to specify "Africa (excl North Africa)" e.g in result section "Information on highest completed level of education in Finland or abroad, economic activity, and Finnish/Swedish language skills (official languages of Finland) was gathered during the survey." Are these all self-reported? Or drawn from other sources? Discussion Page 14, line 51-53: "In Russia and Estonia, adherence to preventive measures has been reportedly influence by the authorities' ambivalent and dismissive attitudes and the citizens' perceived lack of support" should it be "influenced"? and "citizens`" Digital competencies could be mentioned as a reason for the inverse association between older age and downloading the app. It would have been interesting to also know from which sources (and to which degree) immigrants seek COVID-related information. If it is from web-pages and news from their home country, the information received may differ from advice given by the Finish authorities. Moreover, the association with Finish/Swedish language proficiency may be different for immigrants for whom important information is translated to their native language and for immigrant who are dependent on information in Finish/Swedish. This could be discussed, and if data is available also included in analyses. It would also be good with an indication to which groups are covered by the initiatives from the authorities to translate and disseminate information. The authors could also have set the context a bit around barriers to follow advice in various groups; e.g. some groups of immigrants have less permanent attachment to work life and employers and sickness absence may have larger consequences than for others, many work in occupations with more contact with other people making it more difficult to keep distance, but perhaps more necessary to use face masks. Further, social expectations from friends and family may vary between groups. Limitations: Although the authors have implemented several means to include as broad as possible from immigrant populations, including translations and telephone interviews, chances are high
--	--

	that immigrant with poor Finnish/Swedish proficiency, low levels of literacy and not covered by the largest immigrant language translations are less likely than other to participate, and probably also least likely to perceive COVID information as available. This should be discussed. Moreover, in general; people not interested in gaining information on COVID-19 or health related issues, or in following advice would probably not take part in a survey about the topic. I would guess numbers for perceived available information and adherence to advice is somewhat higher than the reality.
--	--

REVIEWER	Behrouz Nezafat Maldonado Imperial College London
REVIEW RETURNED	11-Jan-2023

GENERAL COMMENTS	Thank you for the opportunity to review this important piece of work. The authors used an interesting approach to be able to signal the differences between local and migrant population when it comes to COVID-19 information and prevention measures. They have presented a comprehensive account of their methodology and have reported their results appropriately. Their discussion provides further work to be done as well as policy considerations. I believe this work is worth of publication in view of the impact it can have on future pandemic response.
--

VERSION 1 – AUTHOR RESPONSE

Reviewer: 1

Dr. Jeffry Oktavianus, City University of Hong Kong Comments to the Author:

Thank you for the opportunity to read the article comparing the access to COVID-19 information and adherence to preventive measures between migrant and general populations. The topic is timely and important, especially since migrant communities have been seen as more vulnerable during the health crisis. However, I have some minor recommendations and questions.

1. More background information about Finland and the migrant population will be helpful for the readers and can strengthen the importance of the study.

Thank you for this suggestion. The following brief description about Finland and the migrant origin population was added into the introduction section:

“Finland is a country in North of Europe, with the population of 5.6 million. Migration to Finland rapidly accelerated over the past decades, from 0.5% of the total population in the 1990 to 8% in 2021 (Statistics finland). In 2021, the largest migrant origin populations in Finland were persons born in the former Soviet Union, Estonia, Russian Federation, Sweden, Irak, China, and Somalia.”

2. The questionnaires were available in 18 languages. How did the authors ensure the consistency of the items across languages and that the translation process did not alter the meaning of the survey items?

Thank you for this important point. The translations were made by professional interpreters and have been reviewed by persons speaking the language in question. Additionally, the multilingual interviewers, who were trained in the methodology of conducting surveys, importance of standardization, and why the different questions were being asked, have also reviewed the questionnaires. It is anticipated that these measures reduced the problem of systematic misinterpretations related to translation error or culture-related factors.

3. How did the authors develop the measurement for the perceived access to COVID-19 information? Also, why did the authors choose to measure the perceived access instead of the actual access to the information?

Experts at the Finnish Institute for Health and Welfare (THL) who have an extensive history in conducting population-based surveys and developing survey measures and experts managing the monitoring and reporting on the Covid-19 pandemic developed this measure. All questions in the survey have gone through numerous rounds of comments from highly experienced specialists at THL and have also been tested among volunteer laypersons. Self-perceived access to information was chosen as a measure instead of measured access to information considering the restrictions in length of the survey questionnaire. The limitations of using a self-reported measure were discussed in the limitations section of the manuscript. We also expanded on this important point raised by the reviewer by adding the following text into the limitations section:

“Measured access to information, that may have reduced potential response bias was not included in the current survey due to restrictions in the length of the questionnaire. However, when possible, researchers designing similar surveys in the future are encouraged to consider also including measured access to information.”

4. Page 13, Line 19, should be ‘perceived.’

Thank you, this was revised.

Reviewer: 2

Dr. Marte Kjøllestad, Norwegian Institute of Public Health Comments to the Author:

The association of sociodemographic characteristics with access to Covid-19 information and adherence to preventive measures among migrant origin and general populations in Finland: a cross-sectional study The study assesses self-perceived access to COVID-related information and adherence to preventive measures among immigrants and non-immigrants in Finland. The authors report their survey and the results in an organized and reasonable way, and gives valuable input to the planning of future pandemics and crises. I have only a few comments to consider, outlined below

Methods

Page 5, line 14; it is not clear what is FinMonik?

FinMonik is the abbreviation for the Survey on Wellbeing among Foreign Born Population. The full name of the survey was written out in the text, but the abbreviation “FinMonik” that should have been in brackets following the full name of the survey was accidentally missing. This is now corrected:

“Survey on Wellbeing among Foreign Born Population (FinMonik)”

Line 17: “In addition to including participants in the MigCOVID Survey, a supplementary sample of 982 persons of Somali origin was drawn using simple random sampling and following the same selection criteria as the original sample”. This is not clear, is not the supplementary sample part of MigCOVID/this study?

Thank you for pointing out this typo. It should have been “FinMonik Survey” rather than “MigCOVID Survey”. The FinMonik (conducted 2018-2019) participants were invited to take part in the MigCOVID Survey (conducted 2020-2021). In addition to those who took part in the FinMonik Survey, a supplementary sample of Somali-born individuals was also drawn for the MigCOVID Survey.

The following corrections were made in the text: “In addition to including participants in the FinMonik Survey, a supplementary sample of 982 persons of Somali origin was drawn using simple random sampling and following the same selection criteria as the original sample.”

Russia or the former Soviet Union (FSU); Estonia; the rest of Europe, North America, and Oceania; the Middle East and North Africa; the rest of Africa; Asia or other (later in the text as Asia) What is “or other”? Please specify. In which group are immigrants from Latin America?

The persons from Latin America are indeed in the “other group”. This specification was added into the text. The group size was too small (n=96) to have it a separate group in the analyses of this manuscript.

Suggest to specify that “rest of Africa” will be later called “Africa”, so you do not have to specify “Africa (excl North Africa)” e.g in result section

Thank you for this sound suggestion to make the text more readable. The suggested revision was made in the text.

“Information on highest completed level of education in Finland or abroad, economic activity, and Finnish/Swedish language skills (official languages of Finland) was gathered during the survey.” Are these all self-reported? Or drawn from other sources?

These were self-reported measures gathered during the survey. This was made more explicit in the text: “Information on highest completed level of education in Finland or abroad, economic activity, and Finnish/Swedish language skills (official languages of Finland) was gathered as self-report measures during the survey.”

Page 14, line 51-53: “In Russia and Estonia, adherence to preventive measures has been reportedly influence by the authorities’ ambivalent and dismissive attitudes and the citizens’ perceived lack of support” should it be “influenced”? and “citizens”

Thank you for pointing out these typos. These were corrected in the text.

Digital competencies could be mentioned as a reason for the inverse association between older age and downloading the app.

Thank you for the suggestion to make this specification. It was taken into account:

“Migrant origin populations reported downloading the contact tracing app less frequently than the general population, and the inverse association of older age with downloading the app. This is in consistency previous studies and has been suggested to be related to lower digital competency among older adults[33-34].”

It would have been interesting to also know from which sources (and to which degree) immigrants seek COVID-related information. If it is from web-pages and news from their home country, the information received may differ from advice given by the Finish authorities.

Thank you for this suggestion. This was added into the discussion:

“While majority (91%) of migrant origin persons reported using Finnish media, use of international media (85%) and social media (82%), which may not necessarily convey accurate information of

relevance to the Finnish context, were also frequently reported[16]. Future studies should explore the association of used sources with perceived access to information.”

Moreover, the association with Finnish/Swedish language proficiency may be different for immigrants for whom important information is translated to their native language and for immigrant who are dependent on information in Finish/Swedish. This could be discussed, and if data is available also included in analyses. It would also be good with an indication to which groups are covered by the initiatives from the authorities to translate and disseminate information.

Thank you for this important insight. The issue of the association of perceived access to information with Finnish/Swedish language proficiency among those dependent solely on information in Finnish/Swedish compared with those who also had information available in their mother tongue is important to consider. It is, however, not possible to include this into the analyses because in addition to THL, which produced Covid-19 communications at the national level, also regional authorities and non-governmental organisations produced multilingual communications depending on the population composition/target audience’s needs. It is not fully known, what are all the different languages in which information on Covid-19 was available in Finland.

This important comment was, however, taken into account by specifying in the Introduction section: “In addition to THL that produced crisis communications at the national level, regional health authorities and non-governmental organisations also produced multilingual communications.”

and in the Discussion section: “Those who do not have sufficient Finnish/Swedish skills, and for whom official information on the Finnish context was not available in own language may be at a particular risk of not receiving sufficient information.”

The authors could also have set the context a bit around barriers to follow advice in various groups; e.g. some groups of immigrants have less permanent attachment to work life and employers and sickness absence may have larger consequences than for others, many work in occupations with more contact with other people making it more difficult to keep distance, but perhaps more necessary to use face masks. Further, social expectations from friends and family may vary between groups.

Thank you for this valuable insight. The following was added into the discussion section:

“Working conditions may also be associated with adherence to preventive measures. Less than a third of participants in the MigCOVID Survey who were working or training reported they were able to work remotely, 72% were able to take care of their hand hygiene at work, 56% were able to keep a safety distance to others[16]. Future studies should also explore the association of working conditions, such as ability to work remotely and avoid close contact with others, as well as the association of job security, with adherence to preventive measures.”

Limitations: Although the authors have implemented several means to include as broad as possible from immigrant populations, including translations and telephone interviews, chances are high that immigrant with poor Finish/Swedish proficiency, low levels of literacy and not covered by the largest immigrant language translations are less likely than other to participate, and probably also least likely to perceive COVID information as available. This should be discussed. Moreover, in general; people not interested in gaining information on COVID-19 or health related issues, or in following advice would probably not take part in a survey about the topic. I would guess numbers for perceived available information and adherence to advice is somewhat higher than the reality.

Thank you for highly relevant and important points. These were addressed in the limitations section:

“While considerable measures were taken to increase participation rate, the 18 languages in which the questionnaire was available did not exhaustively cover all of the different languages spoken by the participants. Those who did not participate may have had poorer literacy skills, poorer command of Finnish/Swedish, and may have been likely to speak the languages in which the survey was provided. The languages selected in the survey covered the most commonly spoken languages in Finland. These broadly speaking (but not exactly) corresponded to the languages in which multilingual communications were produced in Finland. Therefore, it is possible that those who did not participate may have been at a greater risk of not receiving sufficient information. Furthermore, those who did not participate may have been less interested in their health and hence less likely to seek information or adhere to preventive measures. Non-response bias was addressed in this study through application of inverse probability weights[23]. These were based on register information on sociodemographics, and for the participants in the FinMonik sample, also earlier survey data on socioeconomic position and type of municipality of residence[4].”

Reviewer: 3

Dr. Behrouz Nezafat Maldonado, Imperial College London Comments to the Author:

Thank you for the opportunity to review this important piece of work. The authors used an interesting approach to be able to signal the differences between local and migrant population when it comes to COVID-19 information and prevention measures. They have presented a comprehensive account of their methodology and have reported their results appropriately. Their discussion provides further work to be done as well as policy considerations. I believe this work is worth of publication in view of the impact it can have on future pandemic response.

VERSION 2 – REVIEW

REVIEWER	Marte Kjøllestad Norwegian Institute of Public Health
REVIEW RETURNED	13-Feb-2023

GENERAL COMMENTS	The association of sociodemographic characteristics with access to Covid-19 information and adherence to preventive measures among migrant origin and general populations in Finland: a cross-sectional study Thanks for the opportunity to read your interesting article again, I think you have addressed the important shortcomings well, congratulations with a nice article! Just a few suggestions/comments included below:  • It could have been good to make clear in the title that access to information was self-perceived access. • Perhaps include a year for the migCOVID survey in the abstract (as quite much happened throughout the pandemic with regard to reaching out with information, it would be useful to know already here whether it reflects the start or the end of the pandemic (or both). • 3368 participants out of 3139 should be a response rate of 55%, not 60%. Moreover, the number does not correspond to table 1; where N=3456. • How did you ensure that participants received information in the right language/the right translated version? Did you have information in all selected languages in the initial invitation letter? • Table 2; try to get each estimate on the same line
--

VERSION 2 – AUTHOR RESPONSE

Reviewer 1, comment 2: The questionnaires were available in 18 languages. How did the authors ensure the consistency of the items across languages and that the translation process did not alter the meaning of the survey items?

Thank you for this important point. A discussion of this issue was added into the Limitations section of the manuscript:

“When conducting surveys in multiple languages, there is a risk that the consistency of the items across the language versions may be jeopardized and that the translation process may alter the meaning of the survey items. In the current study, the translations were made by professional interpreters and have been reviewed by persons speaking the language in question, cross-checking them with the original questionnaire. Additionally, the multilingual interviewers, who were trained in the methodology of conducting surveys, importance of standardization, and why the different questions were being asked, have also reviewed the questionnaires both from the perspective of consistency and conservation of the intended meaning. While it is still possible that some issues remained unnoticed, it is anticipated that these measures considerably reduced the problem of systematic misinterpretations related to translation error or the intended meaning of the items.”

Reviewer: 2

Dr. Marte Kjøllestad, Norwegian Institute of Public Health Comments to the Author:

The association of sociodemographic characteristics with access to Covid-19 information and adherence to preventive measures among migrant origin and general populations in Finland: a cross-sectional study Thanks for the opportunity to read your interesting article again, I think you have addressed the important shortcomings well, congratulations with a nice article!

Just a few suggestions/comments included below:

1. It could have been good to make clear in the title that access to information was self-perceived access.

Thank you for this valid suggestion. This was specified in the title.

2. Perhaps include a year for the migCOVID survey in the abstract (as quite much happened throughout the pandemic with regard to reaching out with information, it would be useful to know already here whether it reflects the start or the end of the pandemic (or both)).

Thank you for the suggestion. The time frame within which the study was conducted was added into the abstract.

3. 3368 participants out of 3139 should be a response rate of 55%, not 60%. Moreover, the number does not correspond to table 1; where N=3456.

Thank you for careful review of the numbers. This was a typo that happened despite careful revision by the authors. The n=3668 (as mentioned originally in the abstract) are all participants in the MigCOVID Survey. The participation rate is 60%. However, since the MigCOVID Survey sample that constituted of persons aged 20-66 had to be restricted to those aged 21-66 years to match the age group of the reference group, n=57 participants in the MigCOVID Survey aged below 21 years were excluded from the current study. This led to the sample of n=3611 participants in the MigCOVID Survey, whose data was used in this study. The group size in the tables 1 and 2 for migrant origin total has now been corrected to n=3611. While there was a typo in this cell in the tables, the numbers by more detailed groups of region of origin in the same table remain correct and they amount to n=3611, as they should. The appropriate revisions were made in the tables 1 and 2 and these were also revised in the manuscript.

In the abstract:

Migrant origin population constitutes of persons born abroad aged 21 to 66 years, who took part in the MigCOVID Survey conducted 10/2020–2/2021 (~~n=3 668~~) (n=3611).

In the methods section:

Altogether ~~3368~~ 3668 persons took part in the MigCOVID Survey, with a participation rate of 60%. Since data for the general population reference group was available only for persons aged 21 years and older, n=57 persons below 21 years were excluded from the MigCOVID participants, resulting in use of data of n=3611 participants in the MigCOVID Survey in the current study.

4. How did you ensure that participants received information in the right language/the right translated version? Did you have information in all selected languages in the initial invitation letter?

Thank you for this question. This was described in more detail in the text of the Methods section. Additionally, we added some points for discussion in the limitations section.

Methods:

“Invitation letters, and paper questionnaires were sent out to the participants in Finnish or Swedish, depending on which of these was marked as the preferred language of communication in the Finnish Population Register, and in their official mother tongue as marked in register. If the questionnaire was not available in the register-based mother tongue of the participant, the post was sent in Finnish/Swedish only. Participants could themselves select the language in which they filled out the electronic questionnaire out of the available 18 languages.”

5. Table 2; try to get each estimate on the same line

Thank you for this formatting suggestion. This was addressed.